# Design, Simulation, and Mechanical Testing of 3D-Printed Titanium Lattice Structures

**Klaudio Bari** 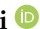

School of Engineering, Computer and Mathematical Sciences, Faculty of Science and Engineering, University of Wolverhampton, Telford TF2 9NT, UK; k.bari@wlv.ac.uk

**Abstract:** Lattice structure topology is a rapidly growing area of research facilitated by developments in additive manufacturing. These low-density structures are particularly promising for their medical applications. However, predicting their performance becomes a challenging factor in their use. In this article, four lattice topologies are explored for their suitability as implants for the replacement of segmental bone defects. The study introduces a unit-cell concept for designing and manufacturing four lattice structures, BCC, FCC, AUX, and ORG, using direct melt laser sintering (DMLS). The elastic modulus was assessed using an axial compression strength test and validated using linear static FEA simulation. The outcomes of the simulation revealed the disparity between the unit cell and the entire lattice in the cases of BCC, FCC, and AUX, while the unit-cell concept of the full lattice structure was successful in ORG. Measurements of energy absorption obtained from the compression testing revealed that the ORG lattice had the highest absorbed energy (350 J) compared with the others. The observed failure modes indicated a sudden collapsing pattern during the compression test in the cases of BCC and FCC designs, while our inspired ORG and AUX lattices outperformed the others in terms of their structural integrity under identical loading conditions.

**Keywords:** DMLS; FEA; failure mode; energy absorption; mesh convergence and divergence

## 1. Introduction

In the field of material science, there have been many recent exciting developments providing new possibilities for engineering solutions, but arguably the simplest and most practical of these advancements are represented by cellular materials. Cellular materials apply the same structural principles as large-scale structures to the mesoscale, creating materials with tailored properties. One common type of cellular material is the truss-based lattice. Innovative titanium models, as shown in Figure 1, were designed and manufactured using a direct metal laser sintering (DMLS) process with powder bed fusion technology, in the School of Engineering at the University of Wolverhampton.

As the density of lattice material decreases so do its mechanical properties, proportionally so, or greater than proportionally [1]. In applications where low weight is required but a certain volume or surface must be filled, cellular structures are ideal. Materials with high specific strength and stiffness, as well as other desirable properties, can be used and their density can be decreased as much as necessary in applications for which they would normally have been too heavy. The use of titanium lattices for bone implants is a perfect example of the benefits of cellular materials. In this case, the biocompatibility of titanium can be deployed, and the high surface-area-to-volume ratio of high-porosity materials enables native bone cells to grow around it to fill all cavities in the lattice structure [2].

*Motivation*

Advances in additive manufacturing and lightweight high-strength materials have allowed unusual and tailor-made orthopedic implants to be constructed. This can ultimately increase the speed of production for bone implants, promote the suitability and

practicality of implants, reduce the risk of rejection, and potentially save lives. This study was specifically aimed at tailoring the stiffness of segmental bone replacement. Many deliverables are anticipated in this study, which includes analysis of two new and innovative lattice designs in comparison with two traditional lattice architectures. Comparisons are a common theme in this study, with the unit cell (UC) and the full-scale lattice comparatively assessed for their merits, a topic into which there has been insufficient research. FEA simulation outcomes were compared with the results of the compression test to assess their validity and accuracy.

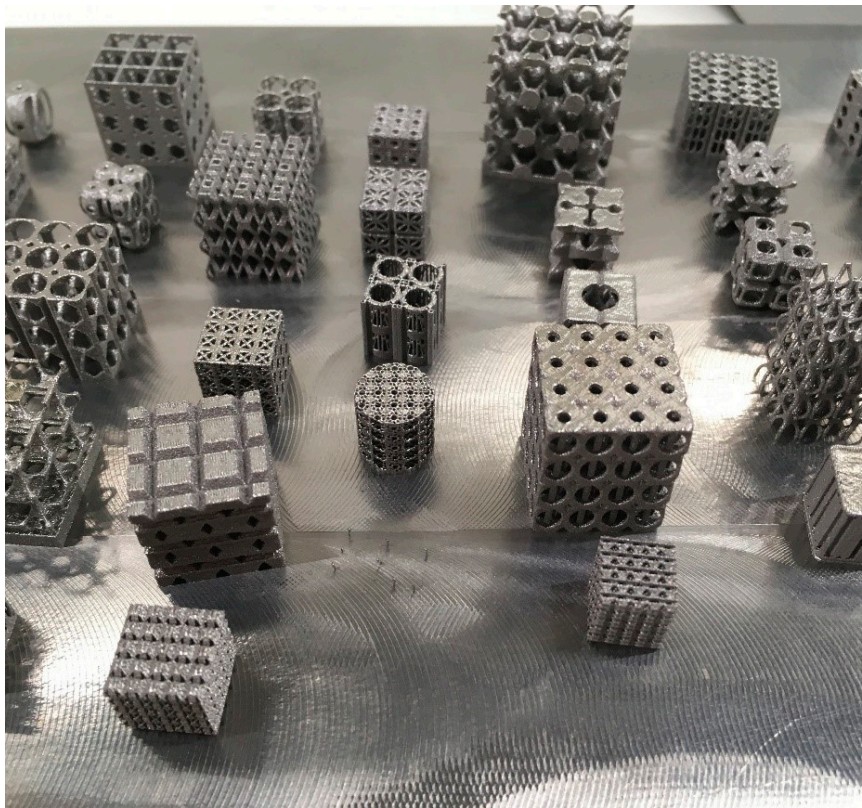

**Figure 1.** Innovative lattice structure showing the truss-based models.

Cellular materials are designed to contain pores on the micro- or mesoscale. These can be classically categorized as foams, honeycombs, or lattices. Another important classification method refers to the stochastic or periodic regularity of cavities within the object. Stochastic closed porosity is naturally occurring and anisotropic, and can be observed in many foam materials on the microscale and macroscale [2]. Uniform periodic cellular materials are formed of equal repeatable cavities. Hierarchical cellular materials, however, are a special class of periodic materials with cavities that vary in shape and size throughout the material according to a predefined pattern. More technically, one can organize these materials by the number of open directions in Cartesian space, as shown in Table 1.

**Table 1.** Classification of cellular material topologies by open direction and uniformity.

| Open Directions | Examples Stochastic | Periodic |
|---|---|---|
| 0, Fully closed | Foam | Pocketed solid |
| 1, Prismatic open | Wood (tracheae) | Honeycomb-based material Corrugated panelling |
| 2, Planar open | | Multi-layered, strut-supported sandwich structure |
| 3, Fully open | Natural sponge Cancellous bone | Lattice structure |

This topology categorization works best in terms of Euclidean geometry for regular third-order polyhedral or cubic shapes. Topology categorization is specific to the internal geometry of the cellular material being used. In this study, the designed materials were open periodic cellular lattice materials, although future work should expand into hierarchical materials.

Comparisons between the different types of cellular materials have been made in various previous studies [3,4] assessing thermal, damping, and mechanical differences as well as common methods of manufacture. The current study mainly considered prismatic topologies, such as honeycombs and corrugated paneling, and their thermal and ballistic properties. When designed correctly, hierarchical cellular materials can offer significant improvements compared with uniform periodic materials [5].

## 2. Methodology

The important properties of cellular materials include compression strength, microporosity, surface roughness, and biocompatibility. However, cellular materials can also be viewed as bulk materials, and their micromechanical behavior measured, including bulk strain, bulk strength, and bulk deformation. Various measurement parameters can be used to define a lattice material relative to the base material from which it is formed [6]. The most influential of these is the relative density of the lattice material, which is also the volume fraction of solid material and the inverse of the bulk porosity; see Equation (1).

$$\rho_r = \frac{\rho_l}{\rho_b} = \frac{V_b}{V_l} = \frac{l}{\rho_l} = \frac{V_l}{V_p} \tag{1}$$

where $\rho_l$ = lattice bulk density, $\rho_b$ = base material density, $\rho_r$ = lattice material relative density, $V_b$ = volume of the base material used in the lattice, $V_l$ = volume of the bulk lattice material, $p_l$ = lattice bulk porosity, $V_p$ = volume of pores within the bulk material.

### 2.1. Unit Cells

For periodic cellular materials, the entire structure can be defined by a representative volume element that contains the smallest linearly repeatable geometry, referred to as a UC. By multiplying out this UC, a cellular material of any scale can be constructed. As the relative density remains unchanged, it, stands that the mechanical properties of a lattice are unaffected by the scale or number of UC used in its construction. The deformation behavior of a material and its material stress distribution are hence often approximated from those of a single UC [6].

Typically, a lattice structure is modeled as a bulk material with homogenized properties and a single UC is assessed using FEA to validate the homogenized approach. One of the purposes of this project is to assess the differences in strain behavior between the UC models and lattices comprised of multiple UCs.

### 2.2. Maxwell's Stability Criterion

One of the theories introduced in this present project and used extensively throughout is the application of Maxwell's stability criterion to truss-based lattice materials [2]. By approximating the cell topology, specifically the UC, as a pin-jointed truss, Maxwell's criterion can be applied as follows:

$$M = b - 3j + 6 \tag{2}$$

where b = number of beams, j = number of joints.

Equation (2) illustrates Maxwell's criterion for pin-jointed truss structures [7]. Polyhedral cell joints are considered movable hinges and columns, while walls are considered rigid beams.

The resulting number determines whether the structure is a mechanism, M < 0, a rigid structure, M = 0, or is statically indeterminate, M > 0. This can broadly be applied to classify the lattice topology by the deformation behavior shown in Table 2:

**Table 2.** Classification of lattice deformation behavior by Maxwell's criterion value.

| | Lattice Structure Deformation Behavior | Example Material |
|---|---|---|
| M < 0 | Bending-dominated lattice | Truss lattice<br>Prismatic (perpendicular to the open axis)<br>Foams |
| M = 0 | Stretch-dominated lattice | Truss lattice<br>Prismatic (parallel to the open axis) |
| M > 0 | Stretch-dominated and self-stressed lattice | Truss lattice |

The Young's modulus $E_1$ of the lattice is strongly dependent on the relative density; this relationship is presented as a plot in Figure 2.

$$E_1 \approx E_b \cdot \rho_r^2$$

where $E_b$ = Young's modulus of base material, $\rho_r$ = lattice relative density = $\frac{\text{bulk lattice density}}{\text{base material density}}$.

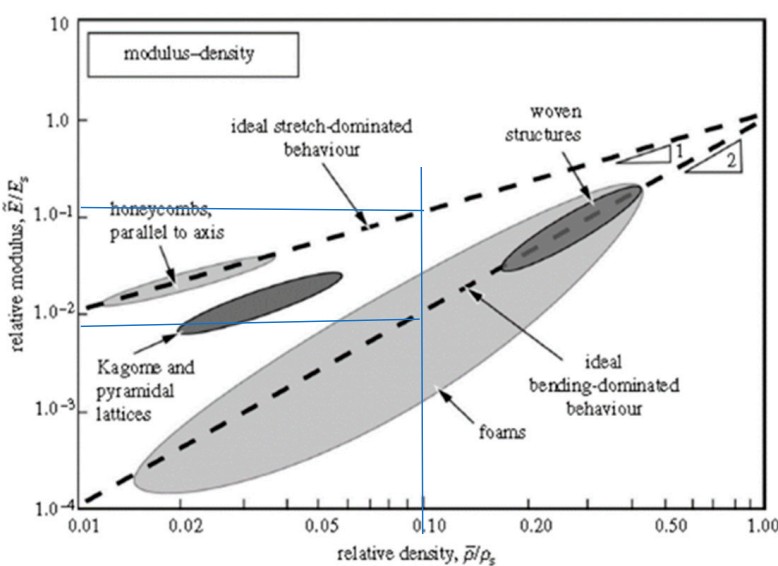

**Figure 2.** Plot of relative density and relative (Young's) modulus for stretch- and bend dominated structures, compared with results from mechanical testing of various cellular materials [2].

Two main concepts were involved in designing the cell units, based on their behavior under axial compression as indicated by the dotted line shown in Figure 2. For example, the relative modulus for a stretch dominated lattice (0.10) is higher than for bending dominated lattices (0.008) at the same relative density (0.1).

### 2.3. Topology Selection

The design principle is based on close simulation of the human femur bone. This extended to topology; hence a design was produced to imitate the stochastic structure of natural bone using a single UC in a repeating pattern [8]. As a control group, two standard strut based Bravais lattices (BCC, FCC) were compared with our inspired AUX and ORG; all four designs are described by acronyms detailed in Table 3. Note that our AUX model was modified from the existing AUX model reported elsewhere, in order to minimize the stress concentration at the edges of the model, making it unique to the current study.

**Table 3.** Summary of acronyms for the four lattice topologies used.

| Type | Description |
|------|-------------|
| BCC | Body-centred cubic |
| FCC | Face-centred cubic |
| ORG | Organic cancellous bone |
| AUX | Auxetic negative Poisson ratio |

All other lattices were designed to exhibit bending dominated collapse behavior. This was achieved to avoid by design any sudden unpredictable collapse and to increase the energy absorbing capabilities during human walking [8].

*2.4. Material Selection*

A lightweight but strong material with biocompatible properties was required for the proposed designs. The titanium alloy Ti6Al4V, often called the 'workhorse' alloy in material science due to its durability, was selected for its proven track record of biocompatibility and desirable shear strength. Ti6Al4V is a two phase alloy containing 91% HPC α-phase and 9% BCC β-phase at room temperature, respectively stabilized by the aluminum and vanadium content [9].

Other materials of interest include Nitinol (NiTi), a shape memory alloy, and CoCrMo, both of which have the potential for common biomedical use in future [10]. A new TiAu alloy has been proposed as an ideal material for lightweight applications, though its antiferromagnetic properties may add complications [10]. However, these materials are not easily manufactured using DMLS, so they were not explored further in the current study. Stainless steel may be another valid and cheaper option, although it has poor biocompatibility and creates other complications during DMLS and bio-integration [11].

Along with most biocompatible materials, Ti6Al4V has almost ten times higher compressive modulus than femur bone (14.8 GPa) [12], hence the bulk Young's modulus (126/14.8 = 0.117) and yield stress require adjustment to avoid stress shielding at the bone–implant interface. The appropriate relative bulk density (0.1) was established from the designated relative bone stiffness as shown in dotted line in Figure 3.

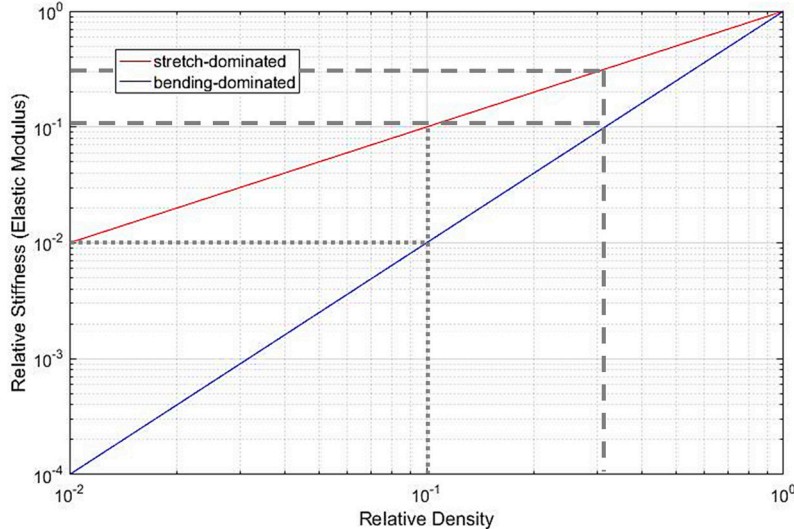

**Figure 3.** Comparison of the theoretical relationship between relative bulk density and relative stiffness (Compressive modulus) for stretch and bending dominated structures.

## 3. Mathematical Calculation

The following calculations were made for the initial design, using the lower bound of combined bone stiffness, 1.4 GPa, as a target [13]. This value was selected to determine

whether the structure would remain structurally sound whilst providing a minimal amount of stiffness.

$$\text{Relative Stiffness} = \frac{E_t}{E_b} = 0.01$$

where $E_t$ = target modulus of cancellous bone= 1.4 GPa, $E_b$ = modulus of lattice base material 126 GPa.

The calculation for relative stiffness of the initial lattice design was:

$$\text{Target Relative bulk Density} = \sqrt{\frac{E_t}{E_b}} = 0.1$$

Hence, the created geometry must be 10% dense and 90% porous. At this same relative density, a stretch-dominated structure would produce a bulk stiffness of 1.4 GPa, so the final values produced must be in this range.

When the design parameters were set, the UC designs were created in CAD using SolidWorks, and the entire lattices were constructed simply by multiplying the UCs by the required amount linearly in both X, Y directions. Based on ISO13314 and the approximate dimensions of a femur bone, a cylindrical shape with height and diameter of 50 mm was used for all lattices.

*Loading and Restraints Setup*

A load of 1140 N was applied during FEA simulation to each of the lattices, calculated as the maximum standing load placed on a single human femur using the method introduced previously [12]. The ground reaction force (GRF) was calculated using the 99th population percentile of bodyweight as follows:

The load on each unit cell (diameter 5 mm) was made proportional, according to the formula below:

$$\text{Load on UC} = \text{GRF} \cdot \frac{A_{uc}}{A_l}$$

where GRF = 1140 N; $A_l$ = CSA of entire lattice = $\pi \cdot (0.025)^2 = 1.9635 \times 10^{-3}$ m$^2$; $A_{uc}$ = CSA of unit cell = $(\text{UC width})^2$.

The recalculation method for UC loading was based on force proportionality. The resulting forces required for each UC were determined, as shown in Table 4.

**Table 4.** Calculation of the load applied on the UC simulation, based on UC widths.

|  | **BCC** | **FCC** | **AUX** | **ORG** |
|---|---|---|---|---|
| UC width (m) | 0.005 | 0.005 | 0.005 | 0.01 |
| Load on UC (N) | 14.64 | 14.64 | 14.64 | 58.569 |

## 4. Manufacturing and Testing

The process employed for additive later manufacturing by Renishaw plc is classified by the ASTM as 'metal powder bed fusion technology', MPBF, which uses high-power laser to sinter the powder for microseconds, and is more commonly known as SLM or PMF. In reality, the layer-by-layer additive manufacturing involves a process somewhere between sintering and melting to build the final product [13].

Due to the high reactivity of Ti at high temperatures, the manufacturing process must be conducted in an inert environment. The RenAM 500 series fully sealed vacuum chamber enables Renishaw's patented atmosphere generation system to quickly prepare a high purity argon environment with minimal gas consumption. The combination of vacuum purging and high integrity system sealing also maintains powder quality and increases longevity.

Internal support structures are not possible for any of the lattices, as they would not be removable. Manufacturing the large horizontal panels of the AUX lattice was therefore a

challenge. To overcome this, the Renishaw technician used flip angled techniques to enable a diagonal 3D printing with external supports. A zigzag laser-beam pattern with 40 μm spot size was used in the melt pool to prepare make the final products shown in Figure 4.

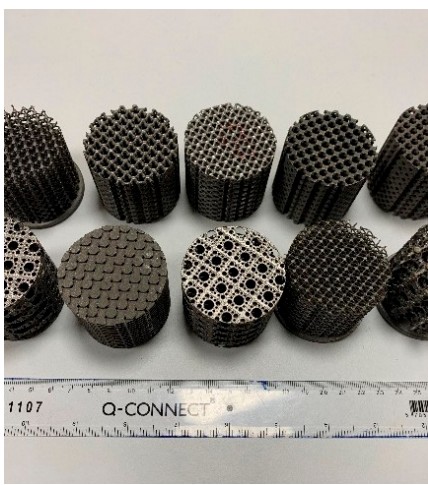

**Figure 4.** Image shows cellular materials produced using the Renishaw Ren AM500 flex machine.

Upon completion, the excess powder was vacuumed away and shaken off the component using an ultrasonic plate or wand. Parts were then removed from the build plate using wire EDM. For samples produced from the second batch onward, additional postprocessing was conducted. Surface finishing was achieved using fine bead blasting with 10 to 50 μm particles, and the following heat treatment cycle was applied:

1. Begin vacuum application to prevent oxidation.
2. Raise temperature for one hour up to 350 °C.
3. Hold temperature at 350 °C for half an hour, to enable the release of all organic volatiles from the surface.
4. Raise temperature for one hour up to 850 °C.
5. Hold temperature at 850 °C for one hour to reach even thermal equilibrium on the interior and exterior of the part.
6. Cool the furnace down to 100 °C.
7. Upon reaching 100 °C, stop applying the vacuum to the chamber.

The α-β crystal phase transition temperature for Ti6Al4V is 995 °C, so this heat treatment process affected the morphology of the α-phase with little effect on the pre-existing β-phase grains, causing precipitate hardening to occur [14]. Any heat treatment above this temperature would dissolve the α-phase and coarsen the β-phase.

*Testing Strategy*

To enable microscopy to be conducted, prototypes were sectioned. Ideally, a cutting process that does not induce heat, such as waterjet cutting, for example, would be used to prevent heat damage to the sample. However, a friction disc cutter was used as there was no preferable alternative available. The samples were then examined using optical microscope (Nikon ShuttlePix 400R low/UK) and Jeol JSM-6400 Scanning Electron Microscope, Oxford Inca Energy Incax-sight. Height maps of the specimens' surfaces were captured using 20 nm gold coating to produce fully focused images.

Multiple specimens for each design are necessary to demonstrate good repeatability, and the relevant international standard states a minimum of five must be used per design [12]. Only one specimen was available for each design in the current study, and further testing will need to be conducted in future.

Specimens should be greased to reduce friction between them and the compression rig. As the simulation was conducted under the assumption of fixed plates on the top

and bottom, this method was not employed, aiming instead to prevent the slipping of the samples. The rig that was used contained a circular groove, exactly 50 mm in diameter, allowing the samples to be locked in place when mounted. As well as preventing unwanted slipping, this also located the samples centrally.

The testing apparatus was a Tinus Olsen H50KT/Wolverhampton, controlled by Tinius Olsen Horizon software (UK) via an external PC. Force, position, and time data were exported from the compression tester. Initially, the compression tester was set to progress at a rate of 5 mm/min until a force of 10 N reached as pre loading condition, Following this, the progression rate was automatically changed to 1 mm/min; hence, all position and time data are interchangeable.

To prevent damage to the testing equipment, the software automatically stopped the compression test when a given drop in force was observed. This was automatically set to 10% or more of maximum force, an indication that the material had failed. When testing lattice structures, many small breakages can occur which do not amount to total material failure but do trigger the automatic shutoff of the compression test. To compensate for this, the threshold was set to a 99% drop.

Based on calibration testing conducted previously, the testing rig's elasticity was compensated for using a compensation factor $C_{RE}$ of 9.490740741. A static measurement variability of 1.7 N was noted.

## 5. Results

The lattice topologies selected were created first as UCs, as shown in Figure 5.

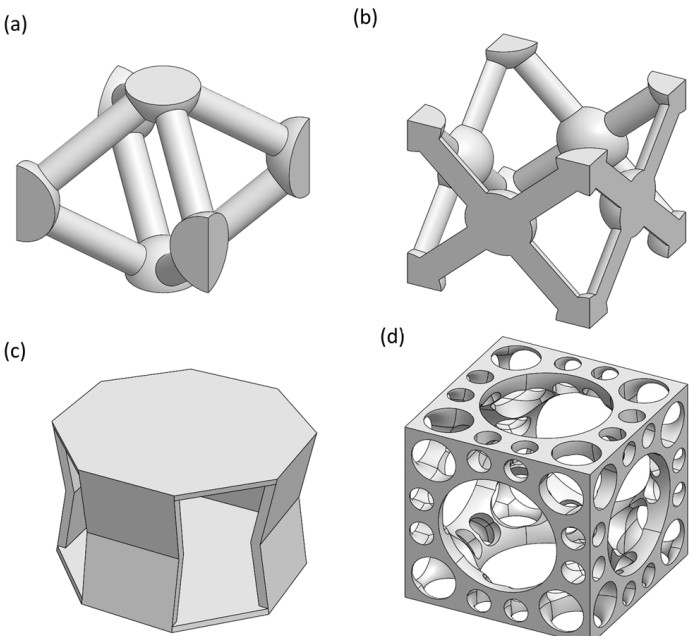

**Figure 5.** UC CAD models for the topologies: (**a**) BCC; (**b**) FCC; (**c**) AUX; (**d**) ORG.

Pin ended truss approximations of these UCs were assessed to confirm that they would exhibit bending dominated deformation, using Maxwell's stability criterion as shown in Figure 6.

By multiplication of these UCs in a linear pattern, lattices were created as shown in Figure 7. From the geometric properties detailed in Table 5, the AUX lattice can be seen to have a much higher surface-area-to-volume ratio. However, the AUX UC was slightly shorter than other UCs and the ORG UC was double the size of the others. To ensure that all lattices had the same density parameters, the weights of all models were fixed between 50–55 g to ensure that no extra materials interfered with the results.

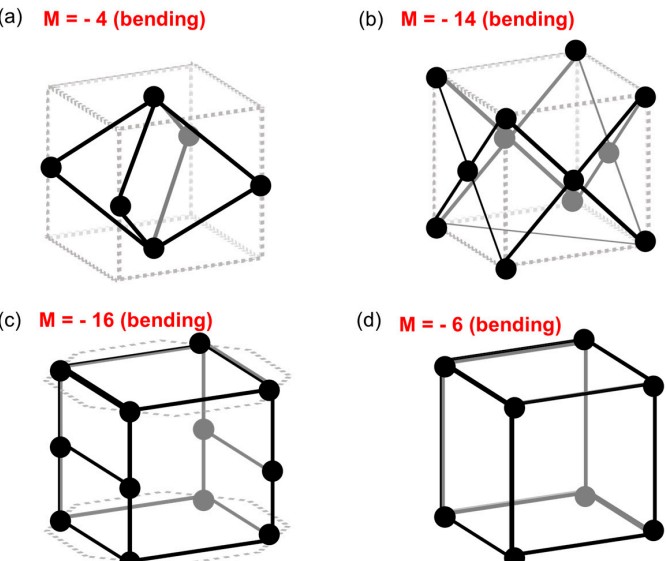

**Figure 6.** Pin ended truss lattice approximations for the topologies: (**a**) BCC; (**b**) FCC; (**c**) AUX; (**d**) ORG. Calculated Maxwell numbers are shown in red.

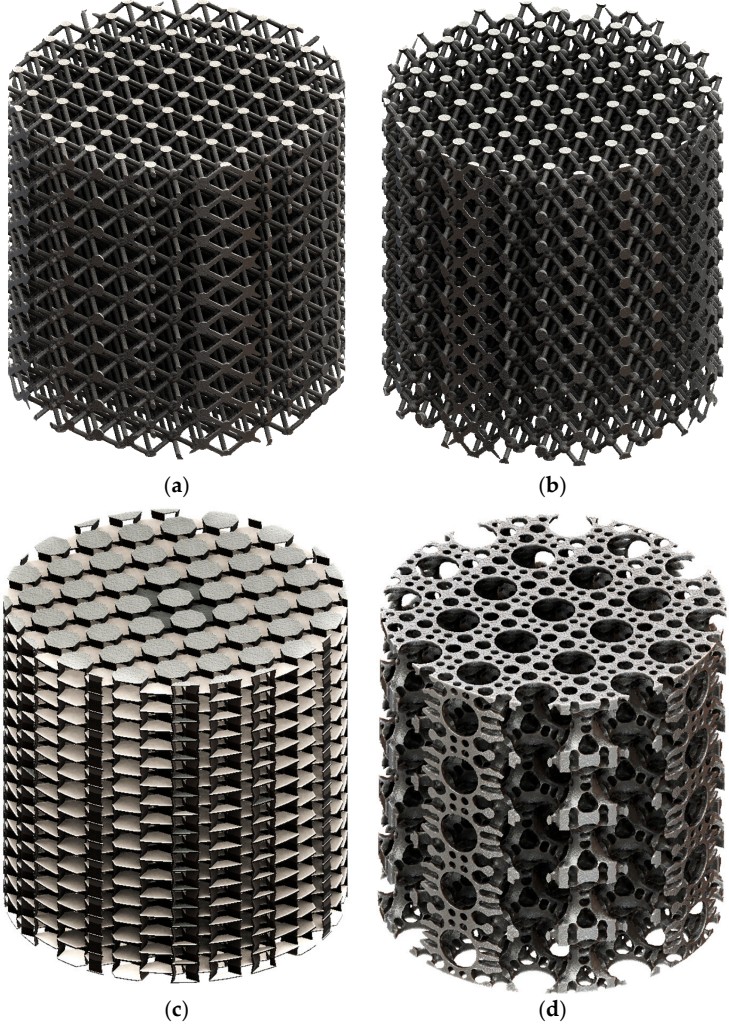

**Figure 7.** Lattice CAD model rendered for the following topologies: (**a**) BCC; (**b**) FCC; (**c**) AUX; (**d**) ORG. Dimension (D: 50 mm, L: 50 mm).

**Table 5.** Summary of geometric properties of lattice models.

| | BCC | FCC | AUX | ORG |
|---|---|---|---|---|
| Total surface area (mm$^2$) | 52,126.8568 | 47,985.4807411 | 104,491.18165 | 44,772.16621 |
| Total volume (cm$^3$) | 392.5 | 392.5 | 392.5 | 392.5 |
| Total mass (g) | 133 | 128 | 138 | 126 |
| Bulk relative density (g/cm$^3$) | 0.338 | 0.325 | 0.351 | 0.32 |
| Porosity | 92.4% | 92.7% | 92.2% | 92.8% |

According to the results of the FEA simulation using the bespoke compression test conditions listed above, the strength of the UC and the bulk lattices' stress distribution are respectively summarized in Tables 6 and 7. The AUX and ORG stiffness values are the closest options to the target stiffness stated above (1.4 GPa).

**Table 6.** Primary results of FEA on UCs, including calculations for bulk stress, strain, and modulus.

| Primary Results | BCC | FCC | AUX | ORG |
|---|---|---|---|---|
| Maximum von Mises stress (MPa) | 234.65 | 48.917 | 51.982 | 130.05 |
| Maximum strain ($\mu\varepsilon$) | 1899.3 | 405.25 | 403.3 | 1011.9 |
| Maximum deformation ($\mu$m) | 6.5481 | 0.65872 | 1.085 | 2.6845 |
| Calculated bulk stress (MPa) | 0.5856 | 0.5856 | 0.5856 | 0.58569 |
| Calculated bulk strain ($\mu\varepsilon$) | 1.3096 | 0.1317 | 0.3100 | 0.2685 |
| Calculated bulk Young's modulus (GPa) | 0.4472 | 4.4450 | 1.8890 | 2.1817 |

**Table 7.** Primary results of FEA on lattices, including calculations for bulk stress, strain, and modulus.

| Primary Results | BCC | FCC | AUX | ORG |
|---|---|---|---|---|
| Maximum von Mises stress (MPa) | 147.1 | 79.511 | 40.485 | 62.952 |
| Maximum strain ($\mu\varepsilon$) | 1204.9 | 642.75 | 315.35 | 522.49 |
| Maximum deformation ($\mu$m) | 34.215 | 11.887 | 6.4167 | 10.265 |
| Calculated bulk stress (MPa) | 0.5806 | 0.5806 | 0.5806 | 0.5806 |
| Calculated bulk strain ($\mu\varepsilon$) | 0.6843 | 0.2377 | 0.1283 | 0.2053 |
| Calculated bulk Young's modulus (GPa) | 0.8485 | 2.4422 | 4.5241 | 2.8280 |

*5.1. Microscopy*

The scanning electron microscopy results showed the proportion of sintering to melting powder particles on the surface of the selected lattice beam. Figure 8 shows the surface of unfused powder particles adhering to beams due to non homogeneous distribution of particle size during the build. Patches of conglomerated metal can be seen but the material is largely sintered and highly porous. The approximate beam diameter in the FCC lattice was 139 $\mu$m.

Manufacturing defects in the AUX lattice can be seen in Figure 9, with sections of the horizontal panels having collapsed. This is a known phenomenon encountered when manufacturing this type of challenging geometry without a support structure and high overhang angle.

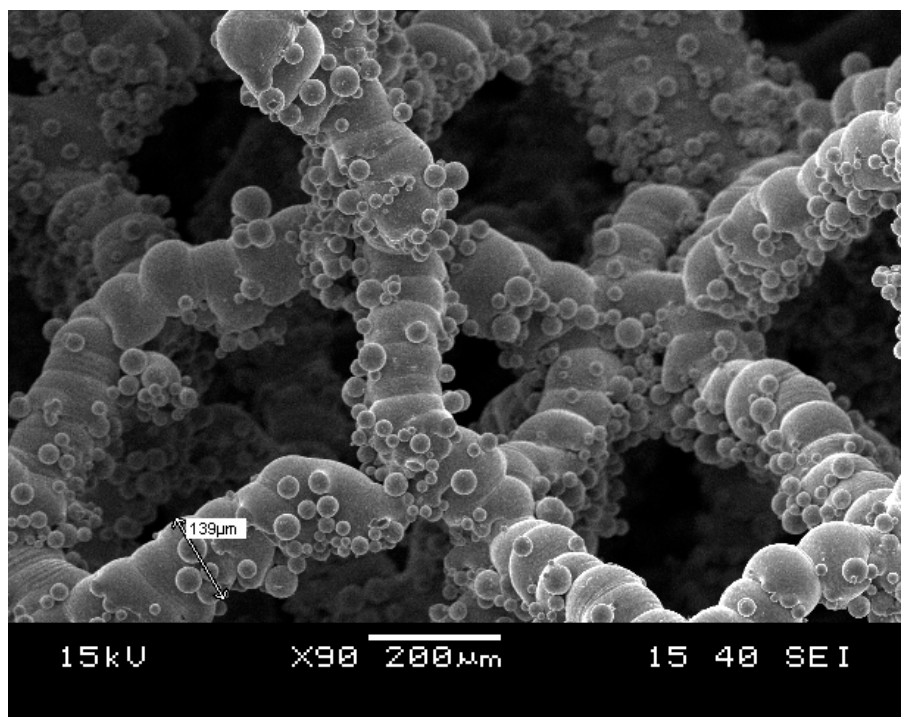

**Figure 8.** SEM image of the micro beams showing the rough surface of the FCC model.

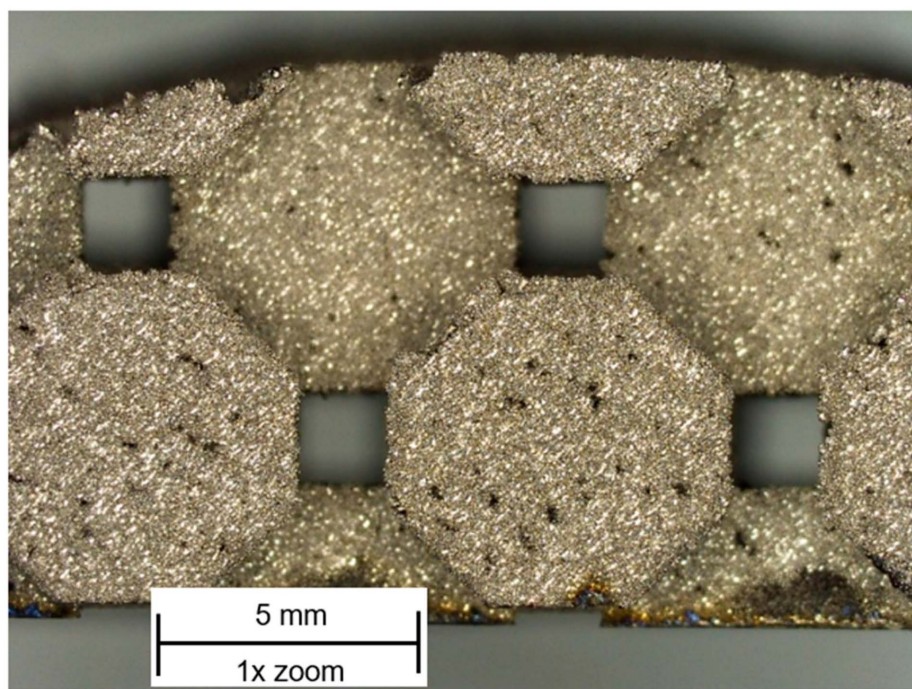

**Figure 9.** Top view of optical macroscope image of AUX lattice section, showing loss of material on horizontal panels.

*5.2. Mechanical Testing*

All four lattices were mechanically compression tested using identical test parameters. The data were exported into an Excel sheet and the results plotted to show the stress–strain curve as presented in Figure 10. The graph clearly shows that ORG, FCC, and BCC had the highest strength values of 25–30 MPa before failure.

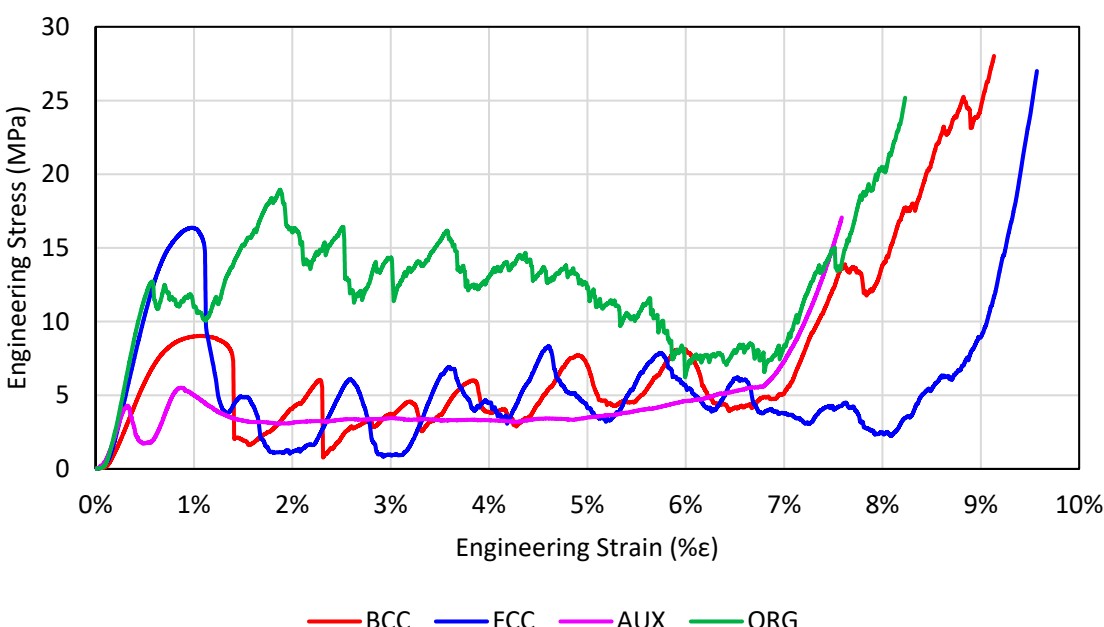

**Figure 10.** Stress–strain curve calculated for the four lattice topologies from the compression test results.

The area under the curve represents the cumulative energy absorbed by the lattice, and it should be noted that the ORG lattice absorbed much more energy (350 J) than the others, as shown in Figure 11.

**Figure 11.** Cumulative energy absorbed during compression testing.

Using linear regression analysis on the elastic region of the compression results, and accepting 99% accuracy or more, the bulk elastic moduli of the test samples were determined as shown in Figure 12 and Table 8 The stiffness of each lattice was within the specified range of the required stiffness, between 1.5–3.0 GPa.

**Table 8.** Summary of Young's moduli for the four lattice topologies from mechanical test results.

|  | BCC | FCC | AUX | ORG |
|---|---|---|---|---|
| Young's Modulus (GPa) | 1.4784 | 2.5881 | 1.9708 | 3.0405 |

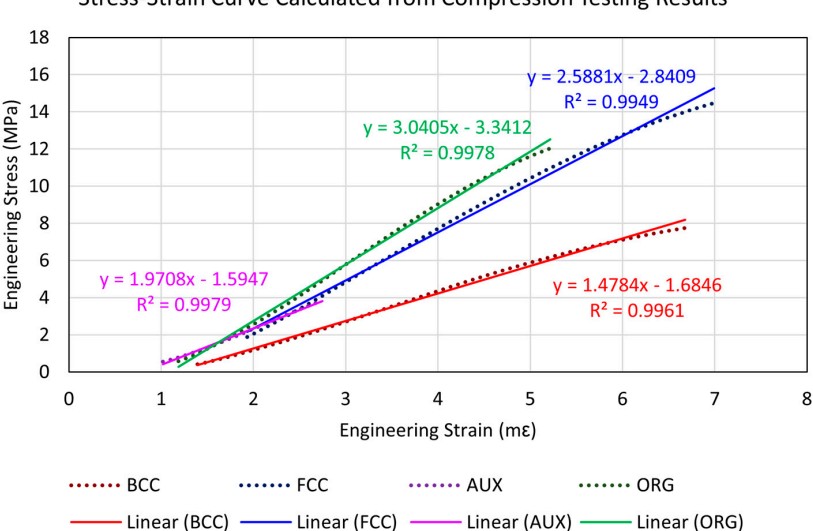

**Figure 12.** Elastic section of stress–strain curve from compression test results. Linear approximations of elastic compression. Solid lines indicate the bulk Young's moduli of the lattices, meeting 99% accuracy or more.

In addition to the quantitative analysis of material properties, the deformation behavior was observed qualitatively. One set of phenomena observed included the quantity and velocity of flying projectiles, seen as debris in Figure 13. The higher densification strain of the BCC and FCC lattices correlated to the amount of material lost during compression.

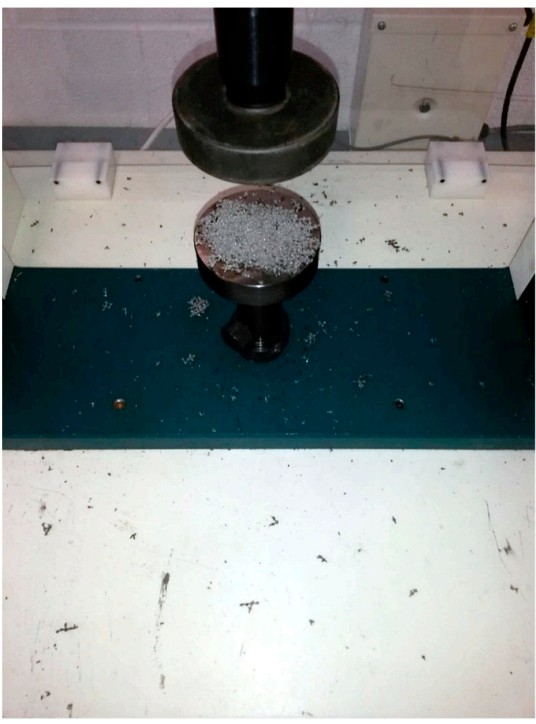

**Figure 13.** The image shows crushed debris from the FCC lattice around machine jigs after compression.

Estimates are provided in Table 9 indicating that the BCC and FCC lattices produced significantly more debris than others during the compression tests. This is a serious concern when considering bone implants inside the body, as these projectiles would risk further damage to surrounding tissue and may block blood vessels around the implant. Meanwhile,

the AUX and ORG lattices were found to be very safe and compact, and collapsed uniformly at their final failing point as shown in Figure 14.

**Table 9.** Summary of estimated material lost as projected debris during compression tests.

|                                | BCC | FCC | AUX | ORG |
| ------------------------------ | --- | --- | --- | --- |
| Estimated material loss (wt%)  | 30  | 35  | 1   | 5   |

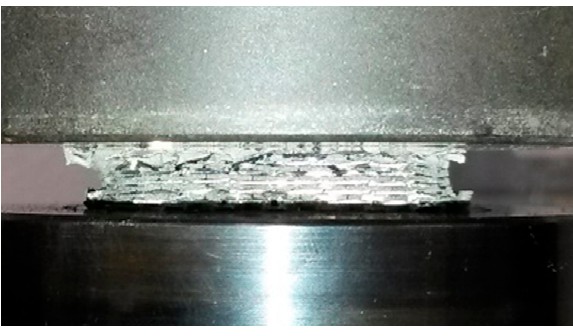

**Figure 14.** Side view of crushed AUX lattice at its final failing point during compression, showing near perfectly uniform collapse properties.

*5.3. FEA Analysis*

In this section, we consider various challenges associated with BCC-UC FEA simulation, such as in Figure 15, which clearly shows that geometric singularities are unavoidable when using tetrahedral elements. A mesh-divergence study on a BCC-UC shown in Figure 16 demonstrates this effect, using a gradually higher number of finite elements in the mesh, resulting in an increase in local stress values on the Y-axis as the number of nodes increases. Hence, we observed logarithmic mesh divergence after 1M nodes, indicating that the meshing is not reliable. Repeating the same process with different mesh shapes (polyhedral, curvature) and/or a different node configuration (Jacobian) did not change the problem of divergence stated above.

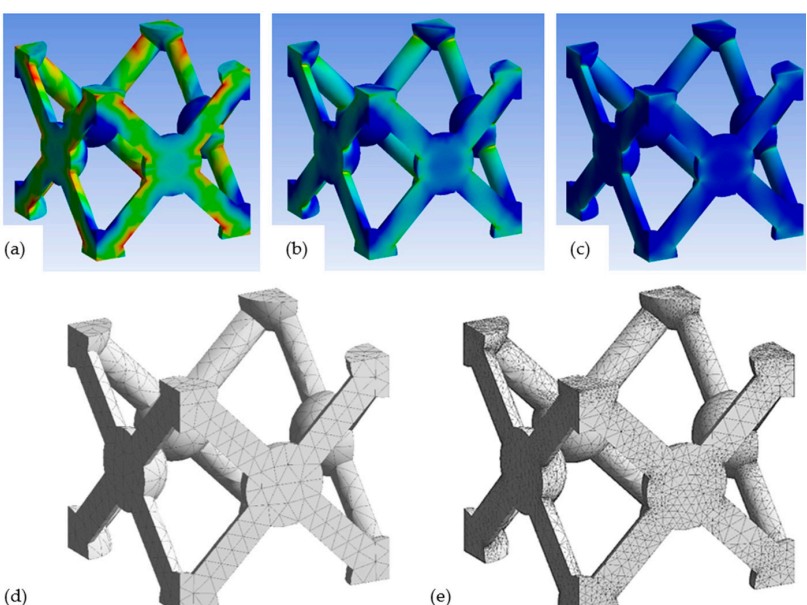

**Figure 15.** Images of BCC UC showing: (**a**) stress results with initial uniform mesh; (**b**) stress results after four mesh iterations; (**c**) stress results after eight iterations (**d**) initial uniform mesh; (**e**) mesh after eight iterative refinements including areas of high stress densely packed with tetrahedral elements smaller than a nanometer.

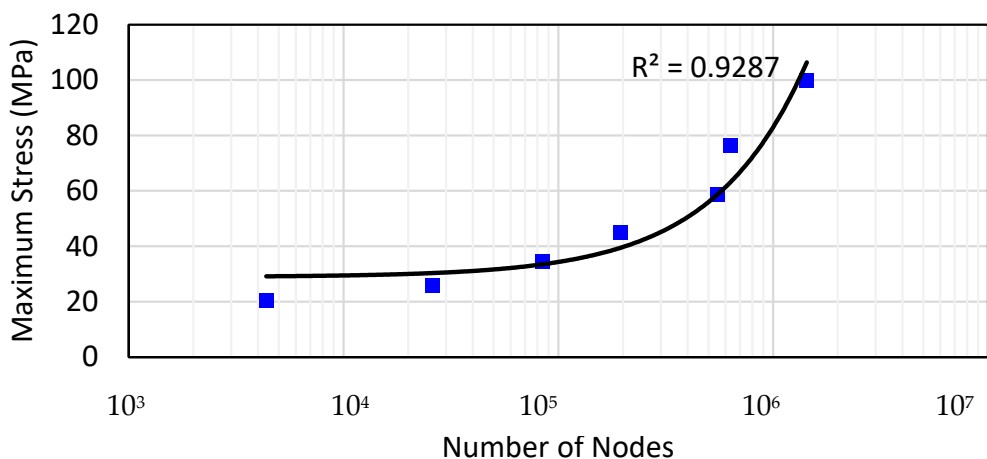

**Figure 16.** Plots of attempted mesh divergence for BCC UC using local stress as the driving parameter.

In comparison to the algorithm for meshing ORG UC structure, a clear independent mesh value of roughly 2.715 µm at 1M nodes is shown in Figure 17. A logarithmic convergence for nodes and maximum deformation can be seen, indicating that the mesh is reliable.

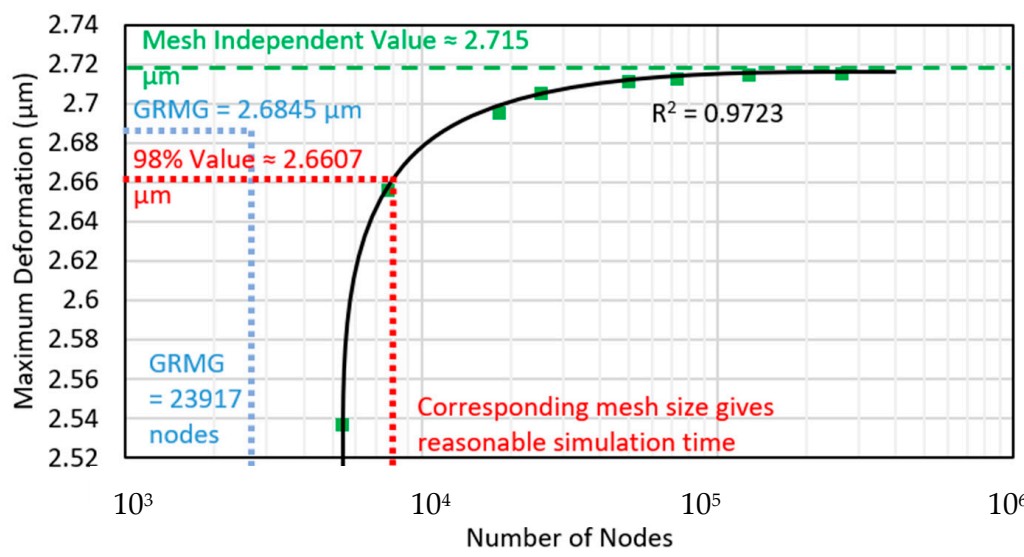

**Figure 17.** Results of mesh convergence for ORG UC using local deformation as the driving parameter.

The results from FEA analysis showed that the maximum deformation observed in the ORG UC was 2.6845 µm, hence this meshing strategy produced over 98% accurate results whilst using fewer nodes than would be required for iterative refinement.

Few obvious errors were found in the simulation results, although a geometric flaw in the initial design for the AUX lattice caused minor anisotropy in results, as seen in Figure 18. This defect is not considered to have had a large impact on the accuracy of the results.

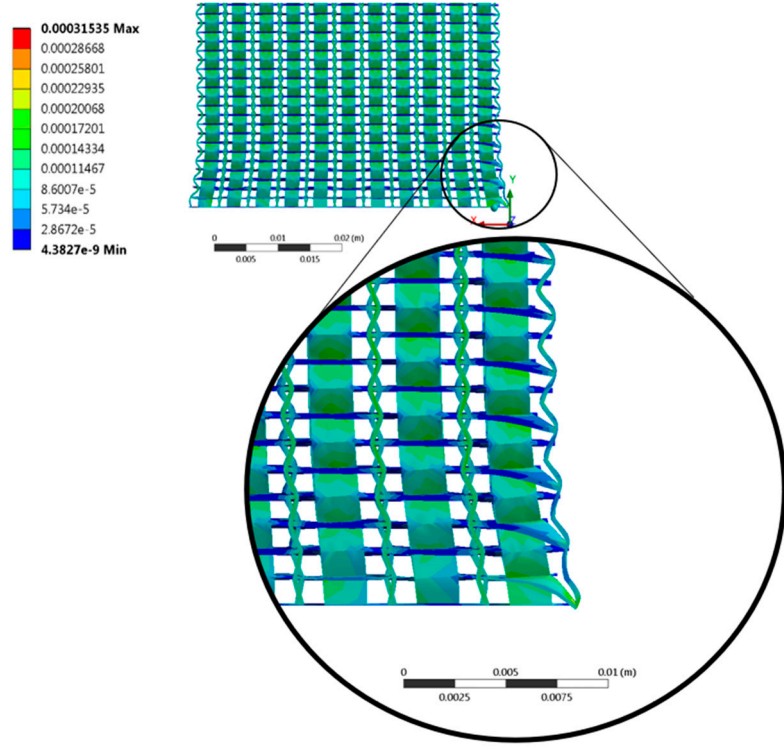

**Figure 18.** Magnified view of strain distribution in AUX lattice at 1000× deformation factor.

The change in stiffness value obtained from UC compared with its equivalent lattice stiffness during design iteration is shown in Figure 19. Considering the elastic modulus, the first design simulations demonstrated significant disagreement in all topologies except for the ORG. Given the evidence described in the previous sections, this was probably due to the presence of singularities and unrepresentative meshing. Furthermore, the results of the first design iteration results reflect a more continuous geometry and are in better agreement.

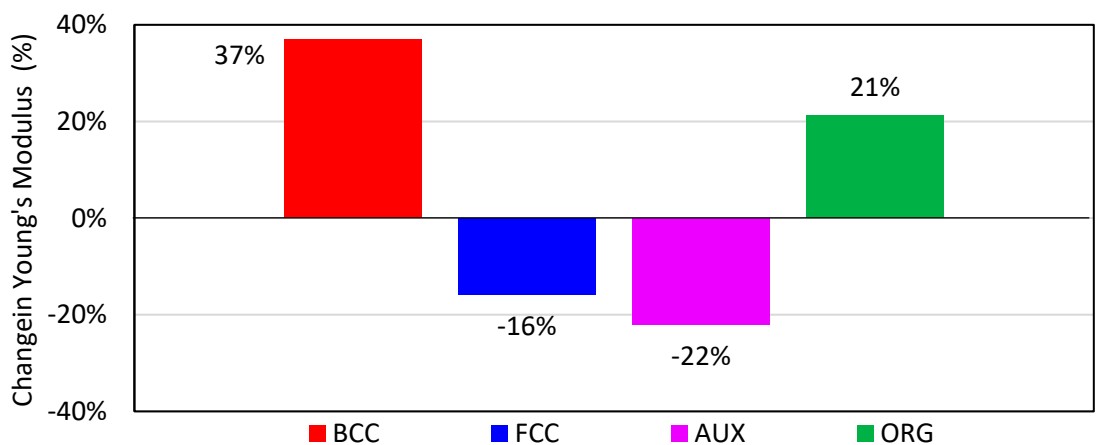

**Figure 19.** Change in bulk Young's modulus from UC to entire lattice simulation for each design.

By summarizing all simulation and experimental results in one plot, the stiffness moduli values from the FEA simulation and the corresponding mechanical test results are compared in Figure 20. The results reveal a discrepancy between BCC and AUX, while ORG and FCC lattices showed good agreement of values. Mesh divergence in the FEA

simulation explains the slight increase in the stiffness values, but the underperformance of mesh divergence in BCC and the excessive mesh divergence in the AUX lattice are evident in Figure 20.

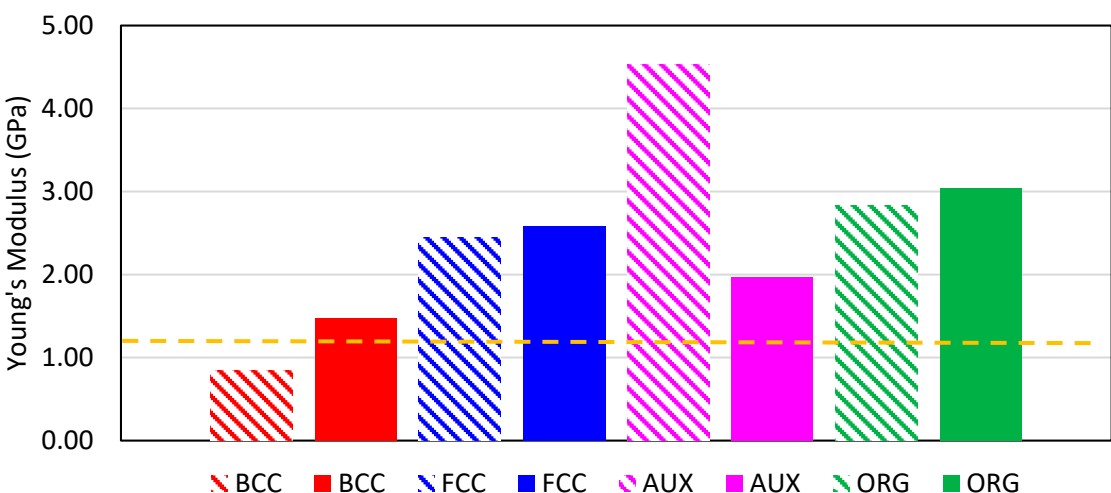

**Figure 20.** Comparison of bulk Young's moduli for the four lattice architectures, determined by simulation (diagonal patterned) and experimental testing (solid colors).

## 6. Fractography

The FEA simulations were conducted well within the elastic range of the compression test; hence, the results do not predict deformation upon plastic collapse, nor was it expected that deformation behavior would be the same for both The plastic collapse of BCC, FCC, AUC and ORG lattice structures are seen in Figures 21–24 respectively. The diagonal red line in the cross-section below indicates why diagonal shear collapse occurred in the BCC and FCC lattices [14].

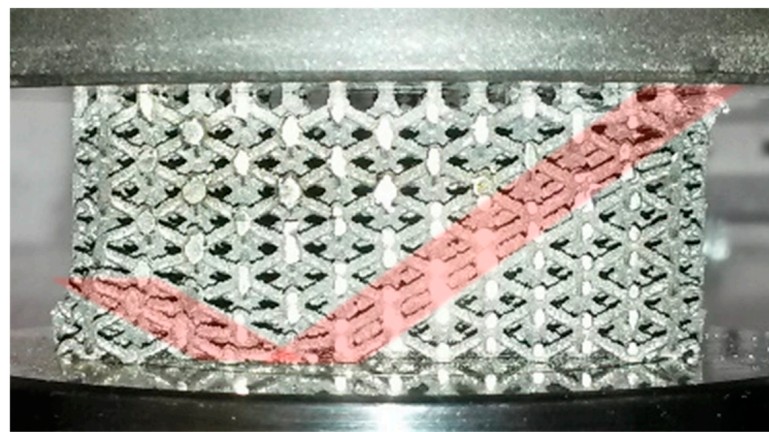

**Figure 21.** Actual deformation in BCC during compression testing. Areas of diagonal shear collapse have been highlighted faintly in red.

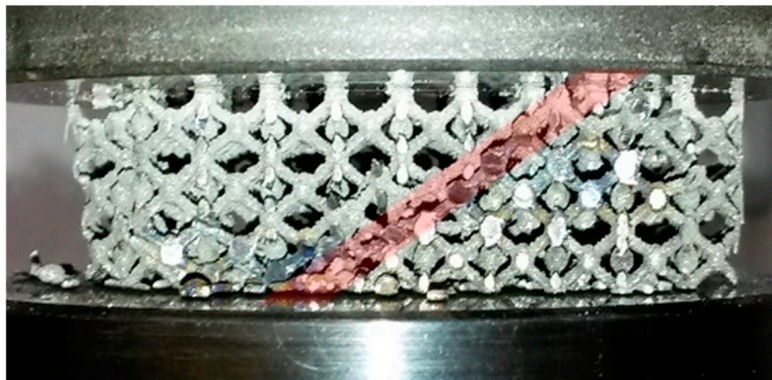

**Figure 22.** Actual deformation in FCC during compression testing. Areas of diagonal shear collapse have been highlighted faintly in red.

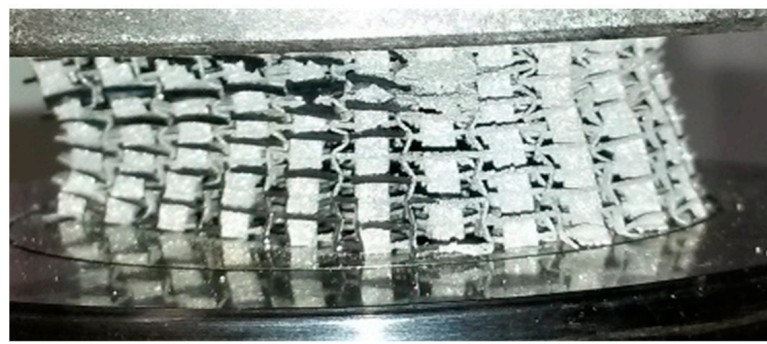

**Figure 23.** Actual deformation in AUX during compression testing. Note the negative Poisson ratio seen at the edges during compression.

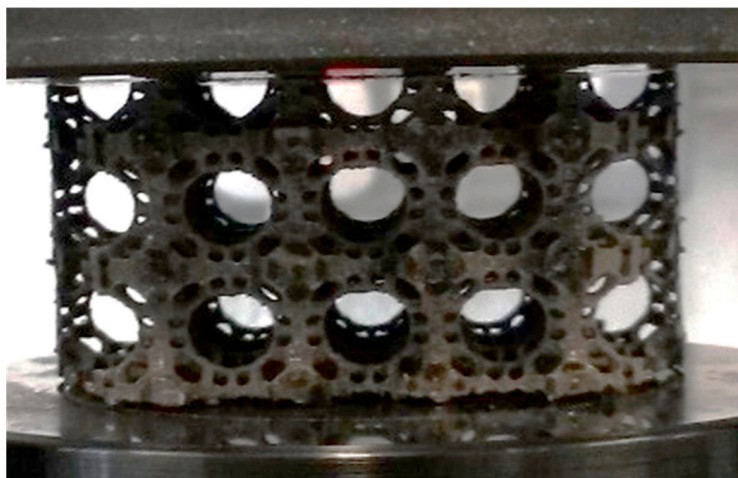

**Figure 24.** Physical deformation profile of ORG during compression testing. Note the uniformity of collapse during compression.

If BCC and FCC were printed not diagonally, their behavior would have more accurately matched that of the simulations. Very little can be seen in common between the elastic and plastic deformation modes, and their failure modes following the diagonal build during 3D printing are shown in Figures 21 and 22. Meanwhile, AUX and ORG were 3D printed vertically, and their failure modes were seen as even collapse, as shown in Figures 23 and 24, respectively.

## 7. Discussion

Additive Layer Manufacturing (ALM) could be used advantageously to connect experts from different disciplines, integrating doctors, design engineers, and manufacturers to reduce the time taken to produce an implant for a patient. With such a method, CT-scan data could be shared instantly between the doctors and engineers who would use CAD to devise the ideal topology, which would be 3D printed by the manufacturer and shipped to the patient.

In vivo, these components must be capable of nucleating and proliferating osteoblasts so that new bone can be grown around the structure. From the microscopy of SLM-produced samples, it is unclear whether the metal powder spheres shown in Figure 8 are too large to allow osteoblasts to attach. It is therefore necessary to conduct extensive further research into various methods of improving surface bonding.

It is proposed here that chemical etching would be a far more cost-effective and logical mechanism by which to introduce micropores onto the lattice surface. Chemical etching is harder to control and produces more variability in results, but as the osteoblasts are also highly varied this should not be an issue. Additionally, the fact that this could be applied via liquid or gas, and not an energy beam, means that every exterior surface could easily be etched evenly. More work needs to be undertaken to determine the necessary level of control and the ideal etching solution or compound. Initial research shows that hydrofluoric, hydrochloric, sulfuric, and phosphoric acids can all break down the oxide layer and etch the material surface. This could then be followed by PEO to restore a strong passivation barrier.

Finally, it is recommended that surface coating via ALD can also be used, which would potentially increase the effectiveness of lattice-to-cell bonding when appropriate micropores are in place. This could be used in place of PEO and surface treatment, as it would improve surface roughness and provide passivation [14].

Tortuosity is another important consideration suggesting that software for stochastic lattice design or the use of modified CT-scan data might be far more effective than a uniform design. A more tortuous path causes osteoblasts to slow down and make more contact with the surface, greatly increasing osteointegration [15].

## 8. Conclusions

This project successfully explored the Maxwell criterion for tailoring the stiffness of titanium bone implants to fit the stiffness target of femur bone, between 1.5–3.0 GPa. The study has highlighted the challenges associated with FEA accuracy during meshing, and compared the results of the compression test for four models. Eventually, when the mechanical properties have been fully assessed, clinical trials and biomedical testing can begin to clear the designs for use as bone implants.

According to FEA, the meshing approach taken for AUX and BCC may have been flawed due to mesh divergence. This could explain some of the discrepancies between the simulation and compression results shown in Figure 20. In future, further mesh convergence analysis should be conducted, although this is only possible in the design of the ORG models. Given the need for large meshes, one possible solution would be to conduct 2D FEA on the lattice cross sections, which would provide useful information while reducing computational times. The accuracy of this approach compared with 3D lattice simulation therefore needs to be assessed.

Hierarchical structures are one of the key areas for future optimization of lattices, and for this application variations in radial density would best replicate the structure of bone. Software such as nTopology with inbuilt hierarchical lattice-design functionalities could be employed for this purpose. Combining this with an image-based or BESO topology optimization program would introduce many more parameters to optimize. Such an optimization tool, hierarchical or otherwise, would save on prototype manufacture and testing time.

Alternatively, a deep-learning algorithm could be employed to create software that mimics the generation and adaption of natural bone. This could allow nature to be represented perfectly, fulfilling the design objective of the ORG lattice proposed in this study.

The failure mode of FCC and BCC raised many concerns related to the debris released at the point of collapse. One of the solutions could be sintering laser-beam control in the DMLS process, which could be used to optimize varying levels of sintering and melting during manufacture. In application to lattice structures, this could be used to ensure that the center of the lattice material is fully melted to ensure good material properties, while the outer surface is sintered to give the required surface roughness.

**Funding:** This research received no external funding.

**Institutional Review Board Statement:** Not applicable.

**Informed Consent Statement:** Not applicable.

**Data Availability Statement:** Data is available on formal request from institutions.

**Conflicts of Interest:** The authors declare no conflict of interest.

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
