# Peer review of "Design, Simulation, and Mechanical Testing of 3D-Printed Titanium Lattice Structures"

_jcs, doi:10.3390/jcs7010032_

Round 1

Author Response

Thank you for the constructive feedback, I have followed the reviewer's advice and re-written the whole manuscript completely from scratch. I do agree with reviewers, it has to be an academic article, not a technical report.

I have consolidated the length of the paper to a reasonable number of pages, highlighting the critical issues in this paper. This is vital to make the article interesting and keep the logic coherent in the text.

Reviewer 2 Report

The author mentions about some project in the Abstract and through the text. At the current stage, the submitted manuscript looks rather like a lengthy project report and not a scientific paper. For the major revision, at least the following should be addressed:

1.   Section numbering is missing. Figure, Eq. and Table numbering should be checked.

2.     Appendices are missing.

3. The Introduction rather follows a textbook pattern. The state-of-the-art overview and novelty of the present paper are missing. Moreover, references are missing for the used tables. Tables 1, 2, 3 and Figures 3, 4, 26 should be removed.

4.  The main part of the manuscript should be reorganized and rewritten. For instance, the initial and improved designs should be presented in the same figure such that the reader is able to compare them immediately. The same holds when the FEA results are compared to Mechanical testing results. This would help the reader to avoid wandering through the manuscript. All unnecessary figures like Figs. 38-41, 43-50, 63-66, 68-75 showing stress or strain distributions should be removed or added as an Appendix.

5.   It is not clear why Figs. 91 and 92 are copied from the other sources and explicitly included in the manuscript. It would be enough just to mention about these works in one sentence.

6.     In the Abstract it is claimed that “… lattices were designed, simulated and mechanically tested for their compressive stiffness and strength.” However, the FE results are only based on the linear analysis. The FE modelling should be extended to include non-linear material response and hence to model the strength of the lattice structures. The simulation results then could be added and compared to the directly exported Force-displacement results from mechanical testing presented in Fig.1 on page 60 (and also in Fig. 55).

Author Response

(The authors gave the same response as above.)

Round 2

Reviewer 1 Report

The paper is acceptable after the authors's careful revising.
